# Isolation and Characterization of Tissue Resident CD29-Positive Progenitor Cells in Livestock to Generate a Three-Dimensional Meat Bud

**DOI:** 10.3390/cells10092499

**Published:** 2021-09-21

**Authors:** Yuna Naraoka, Yo Mabuchi, Yosuke Yoneyama, Eriko Grace Suto, Daisuke Hisamatsu, Mami Ikeda, Risa Ito, Tetsuya Nakamura, Takanori Takebe, Chihiro Akazawa

**Affiliations:** 1Intractable Disease Research Center, Juntendo University School of Medicine, Tokyo 113-8421, Japan; ynaraoka@juntendo.ac.jp (Y.N.); skmdyuvudec6@gmail.com (E.G.S.); d.hisamatsu.ap@juntendo.ac.jp (D.H.); m.ikeda.wa@juntendo.ac.jp (M.I.); r.ito.vh@juntendo.ac.jp (R.I.); 2Department of Biochemistry and Biophysics, Graduate School of Medical and Dental Sciences, Tokyo Medical and Dental University (TMDU), Tokyo 1130-8510, Japan; yomabuchi.bb@tmd.ac.jp; 3Institute of Research, Tokyo Medical and Dental University (TMDU), Tokyo 113-8510, Japan; yone.ior@tmd.ac.jp (Y.Y.); takanori.takebe@cchmc.org (T.T.); 4Department of Research and Development for Organoids, Juntendo University School of Medicine, Tokyo 113-8421, Japan; t.nakamura.sv@juntendo.ac.jp; 5Division of Gastroenterology, Hepatology and Nutrition, Division of Developmental Biology, Center for Stem Cell and Organoid Medicine (CuSTOM), Cincinnati Children Hospital Medical Center, Cincinnati, OH 45229-3039, USA

**Keywords:** CD29, Ha2/5, mesenchymal stem/stromal cells, adipogenic differentiation, culture meat, flow cytometry

## Abstract

The current process of meat production using livestock has significant effects on the global environment, including high emissions of greenhouse gases. In recent years, cultured meat has attracted attention as a way to acquire animal proteins. However, the lack of markers that isolate proliferating cells from bovine tissues and the complex structure of the meat make it difficult to culture meat in a dish. In this study, we screened 246 cell-surface antibodies by fluorescence-activated cell sorting for their capacity to form colonies and their suitability to construct spheroid “meat buds”. CD29+ cells (Ha2/5 clone) have a high potency to form colonies and efficiently proliferate on fibronectin-coated dishes. Furthermore, the meat buds created from CD29+ cells could differentiate into muscle and adipose cells in a three-dimensional structure. The meat buds embedded in the collagen gel proliferated in the matrix and formed large aggregates. Approximately 10 trillion cells can theoretically be obtained from 100 g of bovine tissue by culturing and amplifying them using these methods. The CD29+ cell characteristics of bovine tissue provide insights into the production of meat alternatives in vitro.

## 1. Introduction

Progenitor cells are present in living tissues and are involved in inflammatory reactions and tissue homeostasis [1]. In muscle tissue, satellite cells, known as muscle stem cells, reside beneath the sarcolemma and basement membrane [2]. Muscle contains not only myofibroblasts but also mesenchymal cells that differentiate into fat [3]. Human-derived tissue progenitor cells are being studied worldwide to treat diseases and injuries that cannot be alleviated with current treatments. Tissue-resident progenitor cells are also present in livestock tissues, and meat is being cultivated outside the body as a new field of sustainable development [4].

In recent years, depending on the demand for meat, cultured meat has attracted attention as a means to acquire animal proteins [5,6,7]. Via this method, cells can be obtained by culturing and amplifying proliferative cells from bovine tissue (two-dimensional cell culture). Moreover, Ding et al. verified the expression of Pax7, an undifferentiated marker of muscle satellite cells, in cultured CD29+ CD56+ cells that had been sorted from bovine muscle tissue by fluorescence-activated cell sorting (FACS) [8]. Furthermore, it was revealed that the undifferentiated state of bovine satellite cells can be maintained by suppressing proliferation and differentiation with a p38 inhibitor.

Previously, we developed a technique to isolate proliferative tissue stem cells that exist in mouse and human tissues [9,10]. Platelet-derived growth factor receptor a (Pdgfra)-positive cells are fibroblast-like stem cells present in the bone marrow and muscle tissue and have been identified as the origin of adipose cells [3,11]. However, various kinds of cells reside in tissues, and it is known that their gene expression and cellular properties are heterogeneous [12]. Therefore, the identification and isolation of the progenitor cells that make up tissues are in progress in livestock, but there are problems that must be solved in order to produce meat in this way. First, an efficient method for amplifying muscle cells has not yet been established because muscle cells alone have limited proliferative capacity and require other factors to be present in meat. Second, there are many technical barriers to producing three-dimensional structures with a similar organization to livestock meat.

In this study, we screened cell-surface antibodies (anti-human, anti-mouse, anti-rat, and anti-sheep) using a flow cytometer and assessed their capacity to generate cell colonies. The bovine cell population recognized by CD29 antigen (clone: Ha2/5) was highly proliferative with the capacity to differentiate into muscle and adipose lineages. We also established a culture condition to improve the proliferative capacity of cells using fibronectin. Therefore, we isolated Ha2/5+ cells from bovine skeletal muscle, which have a self-aggregating capacity, to determine whether they are a suitable cell source for the construction of meat-like cell mass “meat buds”. The characteristics of meat buds were examined by analyzing the constituent cells that make up meat to gain insights into the production of meat alternatives in vitro.

## 2. Materials and Methods

### 2.1. Livestock Tissues

The bovine muscle and adipose tissues in this study were derived from two-year-old Japanese black beef (*n* = 8). These meats were obtained from Tokyo Meat Center (Yamanashi, Japan). The pig (*n* = 3) and sheep (*n* = 3) tissues were purchased from Tokyo Meat Center and Charomen Sheep Pasture (Hokkaido, Japan).

### 2.2. Cell Preparation

The cells were isolated from tissue samples as previously described [9,10]. Briefly, the tissues (bovine, sheep, and pig) were mechanically dissected and digested with 2 mg/mL collagenase (#032-22364, FUJIFILM Wako Pure Chemical, Osaka, Japan), 10 mM HEPES (#15630080, Life Technologies, Carlsbad, CA, USA), and 1% penicillin/streptomycin (#15140-122, gibco, New York, NY, USA) prepared in Dulbecco’s modified Eagle medium (DMEM) with GlutaMax (#10569-100, gibco, Bleiswijk, The Netherlands). After shaking for 1 h at 37 °C, the tissue fragments were removed. The supernatant was centrifuged at 800× *g* for 10 min at 20 °C and then collected as mononuclear cell pellets. The cells were washed twice in HBSS and were filtered through a 100-μm cell strainer (#352360, Corning, Durham, NC, USA). The red blood cells in the suspension were lysed by adding 1 mL of ice-cold sterile H_2_O for 6 s. Immediately after lysis, 1 mL of 2 × phosphate-buffered saline (PBS) with 4% fetal bovine serum (FBS) was added to quench the reaction. The cells were then filtered through a 70-μm cell strainer and washed twice. The cell pellet was reconstituted with HBSS (2% fetal bovine serum, FBS) or frozen in CELLBANKER 1 plus (NIPPON ZENYAKU KOGYO CO., LTD., Fukushima, Japan).

### 2.3. Flow Cytometry and Cell Isolation

The cells were resuspended in HBSS and stained with the following antibodies for cell sorting and analysis: allophycocyanin (APC)-conjugated anti-CD29 (anti-rat, clone: Ha2/5, anti-human, clone: TS2/16 and MAR4), anti-CD44 (anti-mouse, clone: IM7), and anti-CD344/Frizzled-4 (anti-human, clone: CH3A4A7) (all antibodies were from BD Biosciences, Franklin Lakes, NJ, USA). For cell surface marker analysis, we screened 246 cell-surface antibodies shown in Appendix A. After antibody incubation on ice, the cells were washed and reconstituted in HBSS. Propidium iodide fluorescence was used to gate out the dead cells. Flow cytometry and cell sorting were performed using a FACS Aria II cell sorter (BD Biosciences).

### 2.4. Cell Culture

The isolated cells were plated on plastic dishes or glass chamber slides coated with fibronectin (#F0635, Sigma-Aldrich MO, USA) in DMEM with GlutaMax (#10569-100, gibco, Bleiswijk, Netherlands) containing 20% FBS (#10270106, gibco, Paisely, UK), 100 units/mL penicillin, 100 μg/mL streptomycin (#15140-122, gibco, New York, NY, USA), and 5 ng/mL basic fibroblast growth factor (#RCHEOT003, ReproCell, Beltsville, MD, USA) in 5% CO_2_ at 37 °C. After cell expansion, the cells were passaged to maintain a density of <80% confluence (passage 2–5 cells used in the experiments). The colony-forming unit fibroblast assay was performed by culturing 2000 sorted cells on a fibronectin-coated 100 mm dish for 14 days in DMEM-GlutaMax containing 20% FBS (#10270106, gibco, Paisely, UK), 1% penicillin/streptomycin, and 5 ng/mL basic fibroblast growth factor (#RCHEOT003, ReproCell, Beltsville, MD, USA). The medium was changed twice weekly.

### 2.5. Generation of Bovine Meat Buds

Cultured CD29+ cells (1 × 10^5^) were seeded in a 96-well U-bottom plate (#4870-800SP, IWAKI, Chiyoda City, Japan) in DMEM with GlutaMax (Life Technologies, Carlsbad, CA, USA) containing 20% FBS (#10270106, gibco, Paisely, UK), 100 units/mL penicillin, 100 μg/mL streptomycin (#15140-122, gibco, New York, NY, USA), and 5 ng/mL basic fibroblast growth factor (ReproCell, Beltsville, MD, USA). The plate was then centrifuged at 400× *g* for 3 min (4 degrees Celsius). The cells were cultured in a state where the cells aggregated at the bottom of the U-bottom wells. A meat bud was formed within 24 h and used in the experiment after culturing for three days. In the case of long-term culture such as differentiation experiments, the medium was changed every three to four days.

### 2.6. Cell Differentiation

In the muscle-differentiation assay, CD29+ cells (or meat buds) were cultured in a differentiation medium consisting of DMEM with GlutaMax (Life Technologies) supplemented with 5% horse serum (#16050122, Gibco, USA), 100 units/mL penicillin, and 100 μg/mL streptomycin (#15140-122, gibco, New York, NY, USA) for five days.

In the lipid accumulation assay, CD29+ cells (or meat buds) were cultured in adipogenic differentiation medium (Lonza, Basel, Switzerland) with 5% horse serum (#16050122, Gibco, New York, NY, USA) and 500 μM oleic acid (#O0180, Tokyo Chemical Industry Co., Ltd., Tokyo, Japan) for two weeks. The adipogenic differentiation medium consisted of adipose induction medium (containing indomethachin, 3-isobutyl-1-methylxanthine (IBMX), insulin, dexamethasone, and L-glutamine: Cat# PT-4136, Lonza, Basel, Switzerland) and maintenance medium (containing insulin and l-glutamine: Cat# PT-4122, Lonza Basel, Switzerland). Two different media were used alternately (every three to four days). Oil Red O staining was performed with cells that had been washed twice with 250 µL of PBS. After 15 min at 25 °C, the Oil Red O solution (#4049-1, Muto Pure Chemicals, Tokyo, Japan) was removed and the cells were washed three times with water. The stained cells were visualized and scanned using a BZX-710 microscope (Keyence, Osaka, Japan) or an LSM 700 confocal microscope (Zeiss, Oberkochen, Germany).

### 2.7. Immunofluorescent Staining

CD29+ cells in the glass chamber were fixed in 4% paraformaldehyde for 10 min. After washing with PBS, the cells were permeabilized with 0.2% Triton-X/PBS for 5 min and blocked with Blocking One solution (#03953-95, Nacalai Tesque, Kyoto, Japan) for 1 h. The following primary antibodies were used: anti-desmin antibody (1:100 dilution, #D1033; Sigma-Aldrich, St Louis, MO, USA), anti-myosin antibody (1:100 dilution, # M1570; Sigma-Aldrich, MO, USA), anti-Pax7 antibody (1:50 dilution, #sc-81648; Santa Cruz, CA, USA), anti-phalloidin antibody (1:100 dilution, #R415; Invitrogen, Carlsbad, CA, USA), and anti-CD29 (Ha2/5) antibody (1:500, #555004; BD Biosciences, Franklin Lakes, NJ, USA). BODIPY^®^ 493/503 was used for lipid staining (Thermo Fisher Scientific, Waltham, MA, USA). The primary antibodies were incubated with 2% BSA/PBS at 4 °C overnight. The cells were washed with PBS and stained with APC-labeled anti-goat mouse IgG (1:1000, Alexa Fluor 594; Abcam, Cambridge, UK) and Hoechst 33342 (#PN226, Dojindo, Kumamoto, Japan). The cells were then washed and mounted using Fluoroshield™ mounting medium (TA-030-FM, Thermo Scientific, Waltham, MA, USA. The stained cells were visualized and scanned using a BZX-710 microscope (Keyence, Osaka, Japan).

For whole mount immunofluorescent staining, CD29+ meat buds were harvested by pipetting and fixed in 4% paraformaldehyde for 24 h. For whole-mount staining, the fixed CD29+ meat buds were washed with PBS, permeabilized with 0.2% Triton-X/PBS for 5 min and blocked with Blocking One solution for 1 h. Primary antibodies (the same as those for normal Immunofluorescent staining) were used with 2% BSA/PBS and 0.1% Triton-X at 4 °C overnight. CD29+ meat buds were then washed with PBS and stained with APC-labeled anti-goat mouse IgG1 (1:1000, #A21125, Alexa Fluor 594, Life Technologies, Austin, TX, USA). CD29+ meat buds were washed and mounted using Fluoroshield mounting medium (TA-030-FM, Thermo Scientific, Waltham, MA, USA). The stained CD29+ meat buds were visualized and scanned using an LSM 700 confocal microscope (Zeiss, Oberkochen, Germany).

### 2.8. RNA Isolation and Quantitative Real-Time PCR

The total RNA was extracted from cells using the RNeasy Plus mini kit according to the manufacturer’s instructions (#74136, Qiagen, Hilden, Germany), and cDNA was synthesized with 100 ng RNA, oligo (dt) primers, and PrimeScript II Reverse Transcriptase (#6210A, Takara, Shiga, Japan). The 200-fold diluted cDNA samples were used for quantitative real-time PCR (95 °C followed by 40 cycles for 15 s at 95 °C and 60 s at 60 °C) in the presence of 10 μM specific forward and reverse primers and Power Track SYBR Green Master Mix (#A46109, Thermo Fisher Scientific, Waltham, MA, USA). Amplification reactions were performed using a Quant Studio 5 Real-Time PCR system (Thermo Fisher Scientific, Waltham, MA, USA). Gene expression was normalized to that of β-actin expression. Each sample was analyzed in duplicate. As a negative control, water was used as the template for each reaction. The primer sequences used for RT-qPCR are described in Table 1.

### 2.9. Analysis of Meat Buds Composition

The samples (muscle tissue, adipose tissue, normal bud, muscle bud, and adipose bud) were harvested and stored at −80 °C. To assess the composition of samples, we used attenuated total reflection-Fourier transform infrared (ATR-FTIR) spectroscopy (IRSptit-T, Shimadzu, Kyoto, Japan). The FT-IR spectra were collected at room temperature using a diamond ATR unit (QATR-S; Shimadzu, crystal diameter of 1.8 mm) in the middle IR region (4000–400 cm^−1^) with a spectral resolution of 4 cm^−1^. To eliminate the influence of moisture on the spectrum as much as possible, the sample was left to stand for several minutes after installation, and the measurement was carried out after drying.

### 2.10. Embedding Meat Buds in Collagen Gel

Cultured bovine Ha2/5+ cells were seeded and cultured in EZ sphere dishes of diameter ~500 µm and depth ~200 µm (#4020-900SP, IWAKI, Tokyo, Japan). A meat bud was used in the experiment after culturing for three days. For embedding meat buds in collagen gel, 1% BSA/PBS (5 mL) was placed into a six-well plate and incubated for blocking (30 min). The pH of collagen (Collagen I, Rat Tail, A10483, Gibco, NC, USA) was adjusted to pH 7.0 on ice. CD29-positive meat buds and collagen were mixed on ice at a ratio of 1:1 and seeded in a six-well plate. To investigate cell apoptosis, meat buds were retrieved from collagen gel with collagenase and analyzed by flow cytometer using Active Caspase-3 Apoptosis Kit (#550480, BD Pharmingen, San Diego, CA, USA).

### 2.11. Statistical Analysis

Quantitative data are presented as mean ± standard error of the mean (SEM) from at least three representative experiments. Statistical analyses were performed using SPSS v26.0 (IBM, Armonk, NY, USA). Statistical analyses on the gene expression were performed by one-way ANOVA with a Bonferroni post hoc analysis for comparison of three or more groups. For comparisons between groups, Student’s *t*-test was used. Results are expressed as mean ± SE. Results with * *p* < 0.05, ** *p* < 0.001 were considered significant.

## 3. Results

### 3.1. Screening for Specific Cell-Surface Antigens on Colony-Forming Cells

Bovine muscle tissue cells were stained with 246 available cell-surface antibodies against humans, mice, rats, and sheep, followed by flow cytometry analysis (Appendix A). The cells were sorted according to positive cells that can be clearly recognized as a cell population (1%≤) and were analyzed for colony formation. In the screening experiment, the presence of a positive fraction in freshly isolated bovine cells was confirmed (Appendix A). The cells of these positive fractions were separated using a flow cytometer, and cell surface antibodies were selected using the colony-forming ability. The antibodies that enriched cells with a high colony-forming ability were CD29 (anti-rat, clone: Ha2/5), CD44 (anti-mouse, clone: IM7), and CD344/Frizzled-4 (anti-human, clone: CH3A4A7) (Appendix A). We examined their colony-forming capacities by comparing Ha2/5+ cells, Ha2/5− cells, and all unpurified living cells (Propidium iodide: PI negative cells) in bovine tissue (Figure 1a). Ha2/5+ cells showed 100-fold enrichment of colonies compared with all living cells (Figure 1b). In addition, the fraction of Ha2/5− cells contained almost no colony-forming cells (Figure 1b). Because CD29 (Ha2/5) is a β1 integrin receptor, CD29 is a beta subunit of the integrin family of molecules that functions as a major receptor for the extracellular matrix such as fibronectin and as an intercellular adhesion molecule [13]. Therefore, we analyzed the proliferative capacity of Ha2/5+ cells on fibronectin-coated dishes (Figure 1c). The cells cultured in fibronectin-coated dishes showed an approximately 23,600-fold increase in proliferative capacity compared with those cultured in dishes without coating (Figure 1c). Interestingly, Ha2/5+ cells were observed as mature cells when the culture was continued (Figure 1d). The Ha2/5+ fraction express CD56 (muscle cell marker: 11.1%) and CD90 (mesenchymal cell marker: 23.8%) (Figure 1e). We also found that Ha2/5+ cells could be isolated using the Ha2/5 clonal antibody in bovine, pigs, and sheep (Appendix A). To investigate the difference in positive rates between Ha2/5 clones and other CD29 clones, we examined the cell surface analysis. The results showed that the positive rate of Ha2/5 clone was higher than that of others (Appendix A). These results indicate that Ha2/5+ cells are present in bovine tissue and that they can form colonies and efficiently proliferate in fibronectin-coated dishes.

### 3.2. Differentiation Capacity to Muscle and Adipose Lineage in Ha2/5 Cells

To analyze the muscle differentiation of bovine cells, we searched for antibodies that could be used for immunostaining (Appendix A). The antibodies and BODIPY available for bovine Myosin, Desmin, and Pax7 were selected by analysis using bovine meat tissue (Appendix A). We induced the differentiation of CD29+ cells to muscle lineage, which resulted in cell fusion (unique phenotype as muscle cells) (Figure 2a,b). CD29+ cells cultured in muscle differentiation medium were immunostained with the muscle differentiation marker Desmin (Figure 2b). To further support these findings, we performed RT-PCR using RNA collected from muscle and adipose tissues as a control (Figure 2c). Ha2/5 cells after muscle differentiation presented increased gene expression of muscle markers (significantly *MYOD* and *MYOSIN*) (Figure 2c). On the other hand, promotion of muscle differentiation reduces *PAX7* gene expression. Interestingly, the presence of oil droplets stained with BODIPY was confirmed in the myotube created through muscle differentiation (Figure 2b). The expression of the lipid markers adiponectin and *LEPTIN* and *ADIPONECTIN* genes was also elevated with 5% horse serum (Figure 2d). The isolated cells from the tissue (passage 0) hardly expressed the muscle gene, but the expression of the muscle-related gene was increased by the induction of differentiation (Appendix A). The results showed that the progenitor cell population contained in Ha2/5-positive cells expresses the myogenic and adipose gene by culture stimulation.

### 3.3. Generation of Bovine Meat Buds from Ha2/5+ Cells

A cell mass with a higher-order structure is formed by self-assembly in a semi-autonomous process [14]. To explore this concept, we examined whether CD29 cells can form spheroids with stem cell properties. First, freshly isolated CD29+ cells and PI-negative cells (unpurified cells) were seeded in U-shaped wells of a 96-well plate and cultured in a muscle cell medium. CD29+ cells had the capacity to form a cell mass by self-assembly (Figure 2a, upper panel). In contrast, the formation of spheroids was not observed in PI-negative cells (Figure 3a, bottom panel). Time-lapse monitoring of CD29+ cells (cultured two passages) showed that cell aggregation occurred within a few hours of culture, leading to the formation of a spherical structure of diameter 0.3–0.5 mm after 24 h (Figure 3b, Appendix A). To investigate the properties of the spheroids derived from Ha2/5+ cells, we examined the effects of normal medium. We found that some of the cells were positive for Pax7, indicating the undifferentiated state of muscle satellite cells (Figure 3c,d). The expression of the muscle differentiation markers was little expressed in the meat buds in DMEM with 20% FBS medium (Figure 3c,d). From the above, it was found that the “meat buds” produced from CD29-Ha2/5 clone-positive cells are muscle cell masses formed by self-assembly with stem cell properties.

### 3.4. Analysis of the Lipid-Accumulation Capacity of Ha2/5+ Bovine Meat Buds

We examined the lipid-accumulation capacity of Ha2/5+ bovine meat buds. Lipogenesis was induced using adipose differentiation medium (containing 500 μM oleic acid) exchanged with fresh medium every three to four days. As a result, we developed a method to efficiently incorporate lipids into bovine Ha2/5 cells (Figure 4a). After 14 days of adipose differentiation induction, the meat buds were found to differentiate into mature adipocytes containing a large number of lipid granules that could be stained with BODIPY (Figure 4b,c). The PCR analysis revealed that the gene expression of adipose differentiation markers (*CEBPα*, *CEBPβ*, *PPARγ*, *LEPTIN*, and *ADIPONECTIN*) was significantly increased in the meat buds with adipose induction (Figure 4d). The expression of the *PAX7* gene was increased by the induction of differentiation (Figure 3e). We compared the amount of muscle, lipid and protein in the meat buds (Appendix A). The analysis using a mass spectrometer indicated the presence of fatty acids with a peak at 1740 cm^−1^ (C=O: Ester) and the presence of protein with peaks at 1650 cm^−1^ (C=O) and 1540 cm^−1^ (N-H, C-H) (Amide I, and Amide II: Amide). The meat bud was close to the protein content of the original bovine meat tissue (Appendix A, Amide). On the other hand, the fat content of the meat bud was far from the adipose tissue (Appendix A, Ester). The presence of fatty acids was increased on the method of inducing adipose differentiation (Appendix A, Ester). Based on these results, meat buds that included muscle contents and could regulate lipid droplets were successfully created by optimizing the induction method.

### 3.5. Formation of Meat Structure Using Bovine Meat Buds and Collagen Gel Matrix

To determine the optimal diameter of culture wells for the formation of meat buds to be embedded in collagen gels, bovine Ha2/5+ cells were seeded and cultured in EZ sphere dishes (Figure 5a). After division, re-seeding, and culture, the meat buds started to proliferate radially, suggesting that the muscle meat buds could be the “seed” that forms the tissue. By incorporating bovine meat buds into the collagen gel matrix, bovine cells could be diffused throughout the collagen tissue (Figure 5b, Appendix A). The meat buds were engrafted in the collagen gel, and after seven days the cell growth became active (Figure 5c). After 10 days, Ha2/5 cells spread throughout the collagen matrix (Figure 5c). These findings show that meat buds incorporated into the collagen were proliferating within the matrix. We analyzed the expression of Annexin V in the meat buds in the collagen gel (Appendix A). The cells were made single with collagenase (1 h) and trypsin (15min) and analyzed using FACS. Although these apoptosis cells include the influence of the process of the experiment, it would not be significantly different for apoptotic cells compared to meat buds in culture.

With our method, we were able to obtain 2.67 ± 0.2 × 10^8^ cells from at least 100 g of meat. Of these, 4.28% ± 1.88% are Ha2/5 positive, and 1.14 × 10^7^ Ha2/5 cells can be isolated. Furthermore, 2.52% of Ha2/5-positive cells presented the ability to form colonies (total 2.87 × 10^5^ colonies). One colony can proliferate into 2.7 × 10^8^ cells after culturing for 21 days. Overall, 7.75 × 10^13^ cells can be theoretically obtained using our isolation/culture method from 100 g of meat (Figure 5d).

## 4. Discussion

In this study, we isolated Ha2/5+ cells from bovine skeletal muscles to obtain cells with proliferative and self-renewal capacity. In addition, we demonstrated that Ha2/5+ cells possess muscle- and adipose-differentiation capacities. Furthermore, the meat buds created from Ha2/5+ cells could differentiate into muscle and adipose tissue when constructed three dimensionally. We found that approximately 10 trillion cells can be theoretically obtained from 100 g of bovine tissue by culturing and amplifying them in fibronectin-coated dishes.

Culturing muscle satellite cells in vitro for a long period has been challenging. Previous reports have shown that muscle satellite cells are activated during in vitro isolation, proceeding to differentiation in skeletal muscle [15]. Our group has established a culture method to maintain muscle satellite cells of skeletal muscle stem cells in an undifferentiated state in vitro [16]. In addition, a technique to self-assemble meat bud tissues in vitro using the liver is being developed [17]. Based on these two fundamental techniques, we created hybrid meat buds using multiple cell populations as meat seeds and established an innovative meat-culture method. By culturing Ha2/5+ cells, Pax7+ cells were maintained for a long period (21 days) and they could proliferate on fibronectin. This suggests that the method is useful for the in vitro culture of muscle satellite cells. In previous studies, CD29 (integrin β1) and CD56 (NCAM) have been used as markers for muscle satellite cells. Choi et al. reported that sorting porcine muscle tissue with the CD29 marker increased the proportion of CD56+/CD29+ cells. These authors also reported that CD56 cells had insufficient muscle-differentiation capacity [18]. Sun et al. performed FACS analysis of cell-surface antigens from pig muscle tissue and found that CD29 was expressed in both adipocytes and muscle cells [19]. In the present study, Ha2/5-positive cells were capable of muscle differentiation and lipid accumulation. Ha2/5 is a cell population with special abilities in bovine tissue, and the identification of these cells and the establishment of a long-term culture method are major achievements in this study.

The previous report shows that adipogenesis of mesenchymal progenitor cells in muscle is strongly inhibited by the presence of satellite cell-derived myofibers [3]. These results suggest that interaction between muscle cells and mesenchymal progenitors has a considerable impact on muscle homeostasis. The CD29 (Ha2/5)-positive cells are a heterogeneous population, and the satellite cells may be limited in number (Figure 1d,e). On the contrary, fat gene expression was elevated when the cells were cultured in 5% horse serum (Figure 2d). This phenomenon may be related to adipose progenitor cells contained in Ha2/5-positive or mature muscle cells. In fact, the muscle cells and adipocytes were observed by continuing the culture without inducing differentiation (Figure 1d). Interestingly, the “dual induction” with adipose differentiation medium and 5% horse serum was confirmed to promote differentiation into fat and muscle (Figure 4). These are unexpected results, which may be due to the properties of Ha2/5 cells. These results showed that the progenitor cell population contained in Ha2/5-positive cells expresses the muscle and adipose gene by culture stimulation.

Currently, our technique is at the laboratory level and is limited to a small scale. Thus, the technique needs to be developed (scaled up) for future mass cultures. If muscle satellite cells cannot be cultured alone, then a mass culture method for muscle satellite cells may be established by adding mesenchymal stem cells, which are known to support the survival and proliferation of muscle satellite cells. However, there is a limitation with regard to production costs. The use of bovine-induced pluripotent stem cells, which are stem cells capable of proliferating indefinitely, may be advantageous in terms of future costs, but the notion of edible induced pluripotent stem cells is not currently socially acceptable. As meat is not the only protein source today, we cannot evaluate whether this technique is superior to protein obtained from livestock only from the viewpoint of environmental effect and safety. For the liver tissue, a “mini-multi-organ” that can generate multiple cells (tissue) simultaneously has been successfully created, which was shown to have a high additional value that cannot be obtained with mixed constituent cells [20]. If meat buds with freely adjustable cell components and nutrients can be created, then meat with a higher additional value to stimulate social demand could be developed.

## 5. Conclusions

Isolation of cells from bovine tissue using the CD29 (Ha2/5) antibody was useful for extracting cells that could become muscle and fat. CD29-positive cells could be engrafted on a collagen-coated dish and could be cultured and amplified in vitro. These cells had the ability to aggregate cells and form meat buds, which are the source of cultured meat.

## Figures and Tables

**Figure 1 cells-10-02499-f001:**
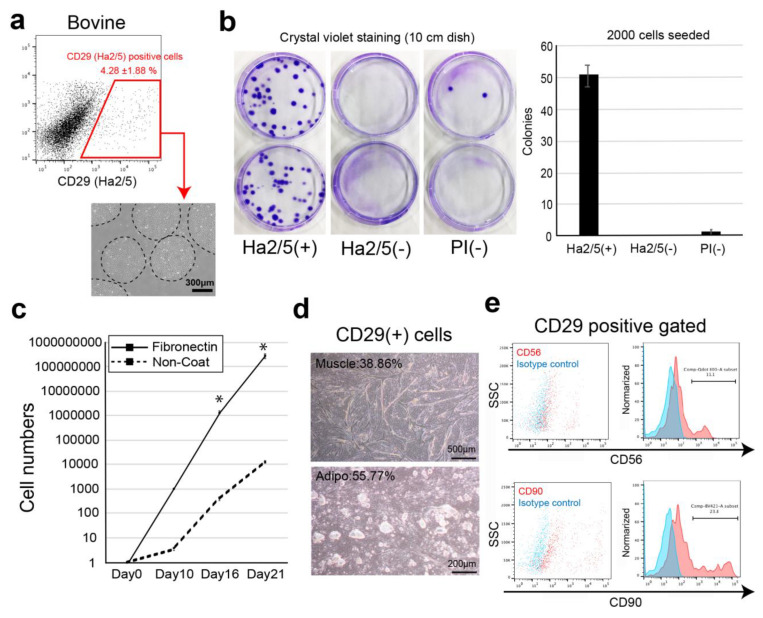
Isolation of colony-forming cells by cell surface markers. (**a**) Representative flow-cytometric profiles of bovine muscle tissue stained with CD29 (Ha2/5 clone) antibody. Phase-contrast micrographs of Ha2/5 positive cells. Scale bar = 300 μm. (**b**) Analysis of colony-forming capacity of the Ha2/5+ fraction. Two thousand CD29+, CD29–, and all living cells were sorted and cultured in 10 cm dishes for 14 days. The cells were stained with crystal violet, and the colonies were counted. (**c**) CD29+ cells were cultured on fibronectin-coated (line) and non-coated (dot-line) culture dishes. The graph shows the cultured cell numbers during 21 days. Results are expressed as mean ± SE (*n* = 3). * *p* < 0.05. (**d**) Differentiation capacity of CD29+ cells. Scale bar = 500 μm. (**e**) The expression of CD56 and CD90 cell surface markers in CD29 positive fraction (red). Isotype control is used for negative samples (blue).

**Figure 2 cells-10-02499-f002:**
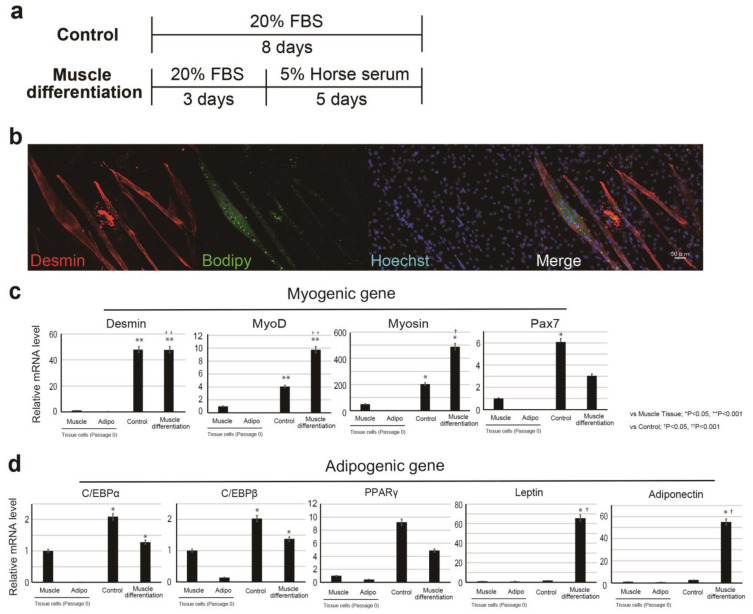
Differentiation of CD29+ cells to myogenic linage. (**a**) Scheme of CD29+ cell differentiation. (**b**) Immunohistochemical analysis of induced cell differentiation. Desmin (a marker of muscle cells) is shown in red; BODIPY (indicating lipid droplets) is shown in green; and Hoechst-stained nuclei are shown in blue. (**c**,**d**) Relative expression of myogenic genes: *DESMIN*, *MYOD*, *MYOSIN*, *PAX7*, and adipogenic genes: *CEBP**α*, *CEBP**β*, *PPAR**γ*, *LEPTIN* and *ADIPONECTIN* in CD29 cells by real-time RT-PCR (muscle: CD29+ cells from muscle tissue cultured without passage, adipo; CD29+ cells from adipo tissue cultured without passage, normal; non-induced cells, muscle differentiation; muscle-induced cells). Using one-way ANOVA with a Bonferroni correction, * *p* < 0.05 or ** *p* < 0.001; compared with muscle cells, ^†^
*p* < 0.05, ^††^
*p* < 0.001; compared with control. (*n* = 4).

**Figure 3 cells-10-02499-f003:**
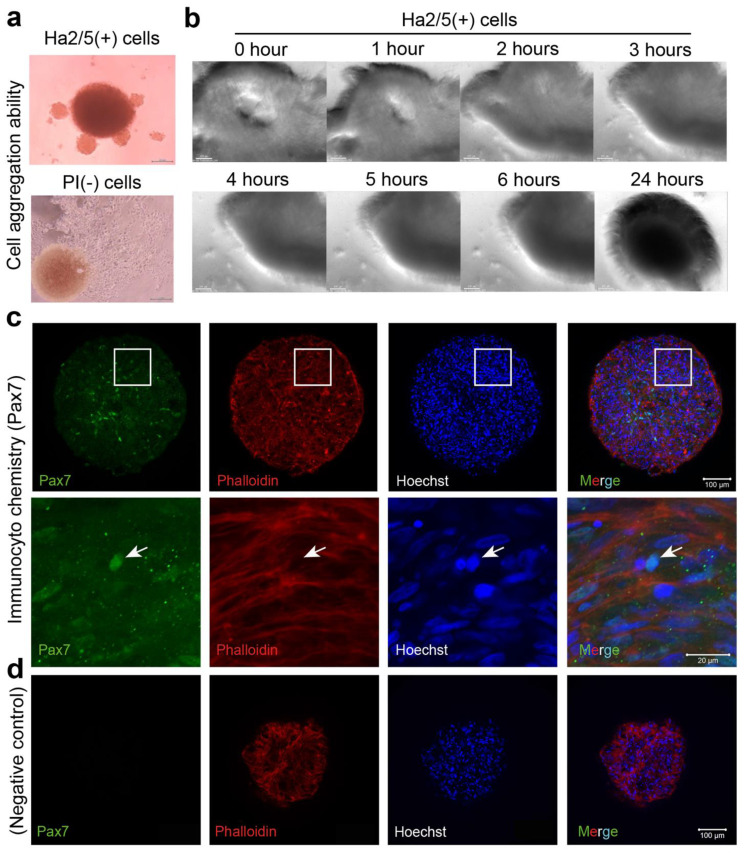
An analysis of the spheroid-forming capacity of Ha2/5+ cells. (**a**) Ha2/5 cells (1 × 10^5^) were seeded in 96-well round-bottom plates, centrifuged at 400× *g*, and then cultured for 24 h. Ha2/5+ cells and unpurified cells (all living cells) were seeded, and photomicrographs taken one week later are shown. The scale bar represents 200 μm. The brown mass (in PI-negative cells) was hematopoietic cells. (**b**) Images of spheroid-forming cells at 0, 1, 2, 3, 4, 5, 6, and 24 h after seeding the cultured Ha2/5 cells. (**c**) Analysis of Ha2/5 spheroids by immunostaining. Muscle marker: Pax7 are shown in green. Cyto-skeletal marker (Phalloidin, red) and cell nuclei (Hoechst, blue) are shown. The white square is enlarged on the lower panel. Allowhead indicate the localization of Pax7. (**d**) Negative controls in Pax7 staining were displayed. The scale bar represents 100 μm.

**Figure 4 cells-10-02499-f004:**
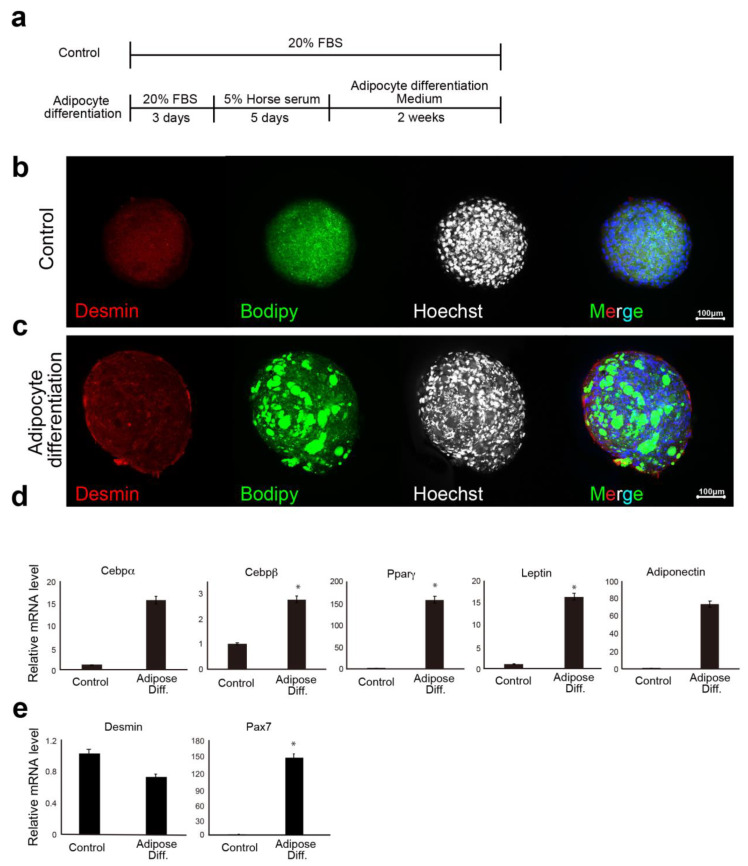
Induction of lipid accumulation and lipid measurement in meat buds. (**a**) Scheme of CD29+ cells with adipogenic differentiation. (**b**,**c**) Spheroids prepared from Ha2/5+ cells were differentiated into adipose differentiation medium for two weeks after five days of differentiation into muscle. Control spheroid (**b**) and spheroids induced to adipose differentiation (**c**) are shown. Desmin (red), BODIPY (green), and Hoechst (Blue). The scale bar represents 100 μm. (**d**) Relative expression of bovine meat bud by real-time RT-PCR. RNA was extracted from adipose-differentiated bovine meat buds (two weeks) for gene expression analysis of adipose differentiation markers (*CEBPα*, *CEBPβ*, *PPARγ*, *LEPTIN*, and *ADIPONECTIN*), and (**e**) muscle markers (*DESMIN* and *PAX7*). (*n* = 4), * *p* < 0.05.

**Figure 5 cells-10-02499-f005:**
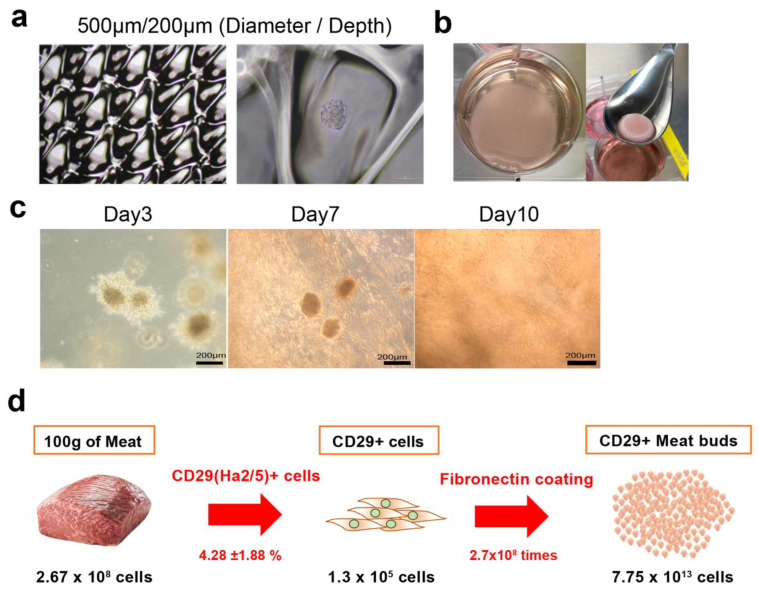
Construction of collagen gel tissue incorporating bovine meat buds. (**a**) Photographs of mass culture of micro-spheroids by spheroid generation plates. Well diameter/depth (500/200). (**b**) Culture of collagen gel incorporating Ha2/5 meat buds. Collagen gel after 24 h (left) and three days (right). (**c**) Phase photographs of the incorporated Ha2/5 meat buds are shown. Time course of collagen gel incorporating Ha2/5 meat buds. Phase photographs of the incorporated spheroids are shown. The scale bar represents 200 μm. (**d**) Calculation of how many meat buds can be made from 100 g of meat in 21 days.

**Table 1 cells-10-02499-t001:** Primer sequences for the RT-qPCR.

	5′-Forward Primer-′3	5′-Revers-′3
*β-ACTIN*	GGCACCCAGCACAATGAAGA	CAGCTAACAGTCCGCCTAGAA
*PAX7*	GACTCCGGACGTGGAGAAAA	TCCAGACGGTTCCCTTTGTC
*DESMIN*	CGGCTACCAGGACAACATTG	CCGGGTCTCAATGGTCTTGA
*C/EBPα*	GCAAAGCCAAGAAGTCCG	GGCTCAGTTGTTCCACCCGCTT
*C/EBPβ*	CGACAGTTGCTCCACCTTCT	CTCGCAGGTCAAGAGCAAG
*PPARγ*	AAATCCCTGTTCCGTGCTGT	GTCAGCTCTTGGGAACGGAA
*LEPTIN*	CGTGACCTTCTTTGGGATTT	AGGGACCATCCACTGAAGTC
*ADIPONECTIN*	CCTGGTGAGAAGGGTGAGAA	GTGACCTGTCTCTCCAGTCC

## Data Availability

The datasets for this study are available from the corresponding author on reasonable request.

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
