# Peer review of "Isolation and Characterization of Tissue Resident CD29-Positive Progenitor Cells in Livestock to Generate a Three-Dimensional Meat Bud"

_cells, 2021, doi:10.3390/cells10092499_

Round 1
Reviewer 1 Report
Introduction
Lines 40-53: Despite the motivation of the study might be the demand for alternative meat production, I believe that this section does not add much to the objectives of the paper. Because to objective of the manuscript is to assess methods to isolate progenitor cells from livestock to be used for production of meat in vitro, I believe the introduction must bring the attention of the author to the constraints or gaps that the trial aimed to fill. Thus, I strongly suggest to the authors to withdraw this section from the introduction.
Line 61: “That is remaining problems…”
Methods
Lines 81-83: The authors have used skeletal muscle tissue isolated from a 2-year-old cattle, is that correct? This sentence is unclear because the concept of meat is different than skeletal muscle. As such, I would suggest to the authors to improve the description of how the tissue isolation procedure was performed to procedure the primary cell culture. This would allow a reader to understand how it was performed and will add robustness of the methods applied making it replicable.
Lines 131-133: How the oil-red-O retained/eluted was used to quantify the lipid accumulation? Absorbance?
Lines 197-199: Statistical analyses are poorly described. I strongly suggest to the authors to improve its description by stating what were used as fixed effects or what statistical model was used in the experiment.
Discussion
Lines 370 – 383: I strongly suggest to delete this section as it does not add much to the discussion of the results obtained and it is a very arguable information.
Author Response
Dear Reviewer:
We thank you for your constructive comments. As you pointed out, we removed inappropriate sentences from the Introduction and added detailed information in the Methods. We think these revisions have improved the manuscript substantially.
Comments of Reviewer #1
Introduction
Lines 40-53: Despite the motivation of the study might be the demand for alternative meat production, I believe that this section does not add much to the objectives of the paper. Because to objective of the manuscript is to assess methods to isolate progenitor cells from livestock to be used for production of meat in vitro, I believe the introduction must bring the attention of the author to the constraints or gaps that the trial aimed to fill. Thus, I strongly suggest to the authors to withdraw this section from the introduction.
Line 61: “That is remaining problems…”
→ We have removed the indicated sentences from the Introduction. Instead, we have provided a background of stem cells and progenitor cells in tissues, and highlighted their potential applications in cellular agriculture (Lines 39–46 and 56–65).
Methods
Lines 81-83: The authors have used skeletal muscle tissue isolated from a 2-year-old cattle, is that correct? This sentence is unclear because the concept of meat is different than skeletal muscle. As such, I would suggest to the authors to improve the description of how the tissue isolation procedure was performed to procedure the primary cell culture. This would allow a reader to understand how it was performed and will add robustness of the methods applied making it replicable.
Lines 131-133: How the oil-red-O retained/eluted was used to quantify the lipid accumulation? Absorbance?
Lines 197-199: Statistical analyses are poorly described. I strongly suggest to the authors to improve its description by stating what were used as fixed effects or what statistical model was used in the experiment.
→ We thank you for the comment. Per your comment, we have revised the Methods. We used the bovine muscle and adipose tissues from 2-year-old Japanese black beef (Lines 77–81). We have added more details on how to extract cells from tissues in the Methods (Lines 83–97). We have revised the statistical analysis section by providing more details (Lines 218-222).
Discussion
Lines 370 – 383: I strongly suggest to delete this section as it does not add much to the discussion of the results obtained and it is a very arguable information.
→ We have deleted indicated sentences from the Discussion. Instead, we have added a sentence regarding the induction of cell differentiation in meat buds (Lines 426-439).
Reviewer 2 Report
This manuscript is excellent! It is very well written, with plenty of detail provided in the methodology to allow replication of methods. Only minor comments:
- the primers used should be included in a table in order to allow replication of the PCR data;
- there should be a separate section in methodology focusing only on the formation of the meat buds as that information is incorporated in the general cell culture, it becomes slightly lost. It should be clearly stated in this section for how long the buds were cultured and in what media;
- the statistical analysis should be included in the figures and also in the figure legends. The authors say that only t tests were used, but for some of the data, t tests are not appropriate. For example, for the data shown in Figure 1b, t tests are not appropriate as 3 conditions are compared, so ANOVA would be suitable. The same for Figure 3g.
- the controls should be clearly stated in methodology;
- how many biological replicates were done should also be clearly specified;
- in the figure legends, it should be specified how is the data shown as. For example, is it mean +/- SD?
- The collagen embedding should be better explained - why is it necessary, what is its scope and how could it be beneficial to cultivated meat?
- It mentions that 10^17 cells could be obtained theoretically from 100g of tissue, but it's not clear how the authors have reached this value. This should be better explained and if calculations were involved, these should be included as data.
Author Response
Dear Reviewer:
We thank you for your valuable comments. We have revised the manuscript accordingly. Our responses to the comments are provided below.
Comments of Reviewer #2
This manuscript is excellent! It is very well written, with plenty of detail provided in the methodology to allow replication of methods. Only minor comments:
the primers used should be included in a table in order to allow replication of the PCR data;
→ We have provided details of the primers used in Table1 (Line 195).
There should be a separate section in methodology focusing only on the formation of the meat buds as that information is incorporated in the general cell culture, it becomes slightly lost. It should be clearly stated in this section for how long the buds were cultured and in what media;
→ As you pointed out, we have added a separate section with the method of meat bud generation in the Methods (Lines 124–133).
the statistical analysis should be included in the figures and also in the figure legends. The authors say that only t tests were used, but for some of the data, t tests are not appropriate. For example, for the data shown in Figure 1b, t tests are not appropriate as 3 conditions are compared, so ANOVA would be suitable. The same for Figure 3g.
the controls should be clearly stated in methodology;
how many biological replicates were done should also be clearly specified;
in the figure legends, it should be specified how is the data shown as. For example, is it mean +/- SD?
→ We added the necessary information in the Figure legends and methods section according to your comment (Lines 218-222, and Figure legends).
The collagen embedding should be better explained - why is it necessary, what is its scope and how could it be beneficial to cultivated meat?
→  Wrapping meat buds with collagen is important to collect meat buds and form large-sized tissues. According to the results of new added experiments, the meat buds embedded in collagen survived and proliferated. These findings suggest that collagen is a useful candidate matrix for collecting meat buds (Figure 6, Lines 344-347).
It mentions that 10^17 cells could be obtained theoretically from 100g of tissue, but it's not clear how the authors have reached this value. This should be better explained and if calculations were involved, these should be included as data.
→ We thank you for your comment. Using our method, we were able to obtain 2.67 ± 0.2 × 108 cells from at least 100 g of meat. Of these, 4.28% ± 1.88 % were Ha2/5 positive, and 1.14 × 107 Ha2/5 cells could be isolated; 2.52% of Ha2/5-positive cells have the ability to form colonies (total 2.87 × 105 colonies). One colony can proliferate into 2.7 × 108 cells after culturing for 21 days. Taken together, 7.75 × 1013 cells can be theoretically obtained using our isolation/culture method from 100 g of meat. The details have been added in (Figure 6, Lines 377-382)
Reviewer 3 Report
1.Figures should be self-explanatory. The abbreviated full name and what is compound 1-6 should be provided in the figure.
2.References for the all methodologies used should be provided.
3.Although the scientific contents are good enough for publication, there still need language improvement. A native speaker is suggested to edit and improve the English presentation.
Author Response
Dear Reviewer:
We thank you for reviewing our manuscript and for providing valuable comment. We have added detailed methodology and cited appropriate references. The revised manuscript was edited by a professional English proofreading service provided.
Comments of Reviewer #3
1.Figures should be self-explanatory. The abbreviated full name and what is compound 1-6 should be provided in the figure.
2.References for the all methodologies used should be provided.
3.Although the scientific contents are good enough for publication, there still need language improvement. A native speaker is suggested to edit and improve the English presentation.
→ We have improved the manuscript per your comments. We have added detailed methodology and cited appropriate references. The revised manuscript was edited by a professional English proofreading service provided.
Reviewer 4 Report
In the manuscript of Naraoke and colleagues ‘Isolation and characterization of tissue resident Ha2/5 positive progenitor cells contained in livestock to generate a three-dimensional meat bud’, the authors were able to isolate proliferative precursors from meat samples. The cell fraction isolated using a CD29 antibody was able to differentiate towards muscle cells and the adipogenic lineage both in 2D and in spheroid culture. Using the CD29 antibody to facilitate cell selection is not new and also not very specific. It is in fact the question if simple pre-plating would achieve the same result. The simultaneous retrieval of muscle and fat precursors (often referred to as fibro-adipogenic precursors) is not novel either. The cells grew in aggregates when cultured in U-shaped wells, as has also been shown previously.
This publication is timely as efficient isolation of satellite cells and adipogenic progenitors is crucial for the field of cultivated meat. The authors do present a novel approach for culturing and differentiating a mixture of apparent satellite cells and adipogenic precursors in a spheroid/aggregate culture.Muscle differentiation in the aggregates is not convincingly shown however and dual differentiation would require more extensive and long term follow up, showing the irreversibility of differentiation when switching to another differentiation stimulus. Even long-term culturing of fat and muscle precursors needs to be tested for equal growth of both cell types as unequal growth will inevitably lead to regression towards a monoculture.
It is an overall well structured manuscript and the data is presented in a clear way.
Major points:
- It is not ideal to use the CD29 antibody clone name Ha2/5 in the title and throughout the manuscript. Using Ha2/5 as a name instead of CD29 is misleading, as it prompts the reader to believe that a new population was discovered, although CD29 was described previously as an option to isolate satellite and adipogenic cells (what the authors did cite correctly). While it makes sense to point out (and could also be discussed in a bit more detail) that not all CD29 antibodies were able to isolate a colony-forming fraction, the correct name of the antigen should be used as the main term (esp. in the title and abstract). It is indeed interesting that the Ha2/5 clone did result in a higher percentage of cells compared to two other CD29 antibodies, therefore a closer look on the isolated cells would be interesting as well, e.g by performing co-stainings during isolation to identify which other potential markers are present.
- The data on muscle differentiation is not convincingly presented.
- In Fig. 1e, representing standard 2D differentiation, myotube formation seems to be a rather rare event (which is compatible with a rather low purity of satellite cells) and Fig.1g indicates that adipogenic markers are more strongly upregulated than myogenic markers, although the differentiation was supposed to favor myogenesis. What is the (rough) percentage of myotubes formed? Can you explain the strong upregulation of adipogenic markers in muscle differentiation? Do you have a differential expression of adipogenic gene expression of cells participating in myotube formation versus the ones that do not? Better to show the freshly isolated muscle and fat tissue as RT-PCR control (as it is of course possible that myogenic or adipogenic genes are highly expressed in the unsorted cells cultured in 20% FBS).
- Fig. 2: It is hard to judge the differentiation from the presented pictures. This comes in part from the low resolution (making the main figures quite blurry), but as well from the low magnification presented, making it hard to judge if the cells are really positive over the strong background. To back up the important conclusion that muscle cells are formed in the meat buds, it would be favorable to add a zoomed-in picture of the formed myotubes and to reduce the background (e.g. by clearing the spheroids). Are cells differentiated with this protocol as well positive for Bodipy? And are Desmin and Bodipy as well co-localised within cells in fresh tissue?
- Negative controls for the antibody stainings should be shown (as supplements) for both fresh tissue and meat buds.
- As the isolated fraction is a mixture of different cells (at least cells from myogenic and adipogenic lineage), staining for additional cell types such as fibroblasts or endothelial cells would be recommended to get a more complete picture.
- The described embedding of spheroids/aggregates in collagen is presumably meant as a preamble to using these aggregates as bioink. Although mentioned that size optimization was performed to prevent cell death, no information is provided, neither of the optimization studies nor of the final construct.
Minor points:
- Figure quality in the manuscript is low and figures seem blurry. Therefore it was hard to judge some of the conclusions drawn
- The material and methods section is a bit short in some points
- line 82: more detailed description of animals used is needed and the formulation is not clear: Was 2 year old meat used or tissue samples of 2 year old animals? How many different samples (and samples from how many different animals) were used?
- Please add how long cells were expanded prior to start of differentiation (how many passages or after how many population doublings)
- Adipogenic differentiation protocol is missing; they only refer to Lonza, which is a general biotech company with many different products.
- The authors gave a theoretical number (line 32, 347) of how many cells can be obtained, but fail to present on how they get this number. Was it calculated from their proliferation data in Fig 1c? Did they test cells after those 21 days for their differentiation potential? Assuming exponential growth and no attrition, it would require 29 doublings to get from the initial number of cells to the final number. For these cells, when cultured in the presence of serum, that is quite a usual number and does not indicate any positive influence of the suggested growth conditions.
- Fig. 4a / line 323 → how was the absence of a necrotic core determined? Please show the data collected
- Supplemental Table: No reference to supplemental tables; reactivity in supplemental table S1 is unexplained. Better call it, for example, % positive cells and specify on which it was tested (fresh meat?)
- Add negative controls for stainings in Supplemental Fig 2
- Typos
- Marge instead of Merge in Fig. 1, 2,3, S2
- Line 260: can form instead of can be formed spheroids
- Line 262: cells were seeded in U-shaped wells of 96 well plate (not in flasks)
- Line 265: culture 2 passages should mean cultured 2 passages?
- Line 282: Phalloidin stained actin and is therefore a cyto-skeletal marker (just using skeletal here is misleading)
- Line 347: the number is not correctly formatted
- Statistical test is mentioned but has not been used apparently. There are no inferences mentioned
- FTIR is not an adequate analysis to qualify muscle protein or fat composition. Interestingly in figure 3 muscle tissue is added as control, but not fat tissue. In view of the more elaborate and successful fat differentiation, using bovine fat tissue as control would be required
- Line 49. This is a mistake. It is not true that only 1% of cow muscle can be used for consumption. The authors must mean something different, but I am not sure what.
- Table 1: phalloidin stain is missing (is mentioned in text). Conversely, reference to anti-myosin antibody in the methods is missing
- Line 221: highly specific: the colony formation assay is mainly showing proliferation and not specific to a certain cell type
- The conclusion is not a conclusion but a methodological summary of the study. The last sentence is more a recommendation than a part of the conclusion.
Author Response
Dear Reviewer:
We thank you for reviewing our manuscript. Through your comment, we understood that Ha2/5 cells contain not only satellite cells but also cells that can differentiate into adipocytes, and their susceptibility to differentiation induction. Per your suggestion, we have performed new experiments and added the relevant details with appropriate controls; this has improved the manuscript. We believe that the development of a method to induce the differentiation of meat buds, which generate muscle and adipose cells, should be studied in detail in the future.
Comments of Reviewer #4
It is an overall well structured manuscript and the data is presented in a clear way.
Major points:
1, It is not ideal to use the CD29 antibody clone name Ha2/5 in the title and throughout the manuscript. Using Ha2/5 as a name instead of CD29 is misleading, as it prompts the reader to believe that a new population was discovered, although CD29 was described previously as an option to isolate satellite and adipogenic cells (what the authors did cite correctly). While it makes sense to point out (and could also be discussed in a bit more detail) that not all CD29 antibodies were able to isolate a colony-forming fraction, the correct name of the antigen should be used as the main term (esp. in the title and abstract). It is indeed interesting that the Ha2/5 clone did result in a higher percentage of cells compared to two other CD29 antibodies, therefore a closer look on the isolated cells would be interesting as well, e.g by performing co-stainings during isolation to identify which other potential markers are present.
→ We thank you for your comment. Following your suggestion, we have named the antibody “CD29” instead of “Ha2/5” (revised the title, abstract, and the main text accordingly). A new experiment was added to investigate the difference in positive rates between Ha2/5 clones and other CD29 clones (Supplementary Figure1c). The results showed that the positive rate of Ha2/5 clone was higher than that of others (Lines 249–252).
2, The data on muscle differentiation is not convincingly presented.
(1) In Fig. 1e, representing standard 2D differentiation, myotube formation seems to be a rather rare event (which is compatible with a rather low purity of satellite cells) and Fig.1g indicates that adipogenic markers are more strongly upregulated than myogenic markers, although the differentiation was supposed to favor myogenesis. What is the (rough) percentage of myotubes formed? Can you explain the strong upregulation of adipogenic markers in muscle differentiation? Do you have a differential expression of adipogenic gene expression of cells participating in myotube formation versus the ones that do not? Better to show the freshly isolated muscle and fat tissue as RT-PCR control (as it is of course possible that myogenic or adipogenic genes are highly expressed in the unsorted cells cultured in 20% FBS).
→ As you pointed out, Ha2/5 culture with 5% hose serum did not strongly induce the differentiation of cells into muscle lineage. The reason is that Ha2/5-positive cells are a heterogeneous population, and the satellite cells may be limited in number (Figure 1d). The satellite cells were concentrated in the CD29+CD56+ fraction (Ding. S, 2018). On the contrary, the fat gene expression was elevated in cells cultured with 5% horse serum (Figure 2d). This phenomenon may be related to the presence of adipose progenitor cells in Ha2/5-positive or mature muscle cells. In fact, muscle cells and adipocytes were observed when the culture was continued without inducing differentiation (Figure 1d). To further support these data, we have added the results of RT-PCR performed using RNA collected from muscle cells (passage 0) as a control (Figure 2, and Supplementary Figure 3). These results showed that the progenitor cell population contained in Ha2/5-positive cells expresses the adipose gene, by culture stimulation with 5% horse serum. We have added the details in the Results and Discussion (Lines 245-247, 426-439).
(2) Fig. 2: It is hard to judge the differentiation from the presented pictures. This comes in part from the low resolution (making the main figures quite blurry), but as well from the low magnification presented, making it hard to judge if the cells are really positive over the strong background. To back up the important conclusion that muscle cells are formed in the meat buds, it would be favorable to add a zoomed-in picture of the formed myotubes and to reduce the background (e.g. by clearing the spheroids). Are cells differentiated with this protocol as well positive for Bodipy? And are Desmin and Bodipy as well co-localised within cells in fresh tissue?
(3) Negative controls for the antibody stainings should be shown (as supplements) for both fresh tissue and meat buds.
→ We have added fine images to show the cellular properties of meat buds (Figure 3). These images are of normal culture meat buds without 5% horse serum. Interestingly, the "dual induction" with adipose differentiation medium and 5% horse serum was confirmed to promote differentiation into fat and muscle (Figure 5). These are unexpected results, which may be due to the properties of Ha2/5 cells. Negative controls for the antibody staining have added in Supplementary Figure 2.
3, As the isolated fraction is a mixture of different cells (at least cells from myogenic and adipogenic lineage), staining for additional cell types such as fibroblasts or endothelial cells would be recommended to get a more complete picture.
→ To investigate CD29-positive cells in more detail, we conducted a cell surface antigen analysis using a flow cytometer. We found that the CD29-positive cells are a mixture of cells expressing mesenchymal/fibroblast markers (CD90 and Pdgfra) and cells expressing muscle satellite cells (CD56) (Figure 1e). The Ha2/5 antibody, which is the focus of this study, recognizes muscle and adipose lineage cells. Therefore, it may be a suitable antibody to efficiently isolate the cells needed to prepare meat buds (Lines 245-247).
4, The described embedding of spheroids/aggregates in collagen is presumably meant as a preamble to using these aggregates as bioink. Although mentioned that size optimization was performed to prevent cell death, no information is provided, neither of the optimization studies nor of the final construct.
→ We looked at the scientific rationale for optimizing cell size and found no significant difference. Therefore, we have deleted the data from the Figure 6. Wrapping meat buds with collagen is important to collect meat buds and form large-sized tissues. According to the results of new added experiment, the meat buds embedded in collagen survived and proliferated. These findings suggest that collagen is a useful candidate matrix for collecting meat buds (line 344-347).
Minor points:
Figure quality in the manuscript is low and figures seem blurry. Therefore it was hard to judge some of the conclusions drawn
The material and methods section is a bit short in some points
line 82: more detailed description of animals used is needed and the formulation is not clear: Was 2 year old meat used or tissue samples of 2 year old animals? How many different samples (and samples from how many different animals) were used?
Please add how long cells were expanded prior to start of differentiation (how many passages or after how many population doublings)
Adipogenic differentiation protocol is missing; they only refer to Lonza, which is a general biotech company with many different products.
→ We thank you for the comment. We have revised the methods section accordingly (Methods section).
The authors gave a theoretical number (line 32, 347) of how many cells can be obtained, but fail to present on how they get this number. Was it calculated from their proliferation data in Fig 1c? Did they test cells after those 21 days for their differentiation potential? Assuming exponential growth and no attrition, it would require 29 doublings to get from the initial number of cells to the final number. For these cells, when cultured in the presence of serum, that is quite a usual number and does not indicate any positive influence of the suggested growth conditions.
→ We thank you for your comment. Using our method, we were able to obtain 2.67 ± 0.2 × 108 cells from at least 100 g of meat. Of these, 4.28% ± 1.88 % were Ha2/5 positive, and 1.14 × 107 Ha2/5 cells could be isolated; 2.52% of Ha2/5-positive cells have the ability to form colonies (total 2.87 × 105 colonies). One colony can proliferate into 2.7 × 108 cells after culturing for 21 days. Taken together, 7.75 × 1013 cells can be theoretically obtained using our isolation/culture method from 100 g of meat. The details have been added in Figure 6 (Lines 377-382).
Fig. 4a / line 323 → how was the absence of a necrotic core determined? Please show the data collected
→  Please see our response to comment 4 (Major points). We conducted a new added experiment and showed that collagen embedded meat buds are alive and proliferating in collagen (Figure 6, Lines 344-347).
Supplemental Table: No reference to supplemental tables; reactivity in supplemental table S1 is unexplained. Better call it, for example, % positive cells and specify on which it was tested (fresh meat?)
Add negative controls for stainings in Supplemental Fig 2
→ Regarding the contents presented in Supplementary Table S1, the relevant methods and results have also been added in the manuscript (Lines 229-233). Immunostaining negative controls have been added in Supplementary Figure 2.
Typos
Marge instead of Merge in Fig. 1, 2,3, S2
Line 260: can form instead of can be formed spheroids
Line 262: cells were seeded in U-shaped wells of 96 well plate (not in flasks)
Line 265: culture 2 passages should mean cultured 2 passages?
Line 282: Phalloidin stained actin and is therefore a cyto-skeletal marker (just using skeletal here is misleading)
Line 347: the number is not correctly formatted
→ We have made the necessary revisions according to your comments.
Statistical test is mentioned but has not been used apparently. There are no inferences mentioned
→ We added the information about statistical analysis in the manuscript (Lines 218-222) and figures.
FTIR is not an adequate analysis to qualify muscle protein or fat composition. Interestingly in figure 3 muscle tissue is added as control, but not fat tissue. In view of the more elaborate and successful fat differentiation, using bovine fat tissue as control would be required
→  As you pointed out, we have added control muscle tissue samples and adipose tissue samples. Regarding the data in Figure 5, the culture conditions with both adipose differentiation medium and 5% hose serum confirmed the effect of promoting adipogenic differentiation. We have added the details in the manuscript (Figure 5, Lines 327-332 ).
Line 49. This is a mistake. It is not true that only 1% of cow muscle can be used for consumption. The authors must mean something different, but I am not sure what.
→ We have revised the introduction section according to your comment. The text that you pointed out has been deleted. We hope that we have adequately addressed your concern.
Table 1: phalloidin stain is missing (is mentioned in text). Conversely, reference to anti-myosin antibody in the methods is missing
→ We have made the necessary revision in the manuscript according to your comment (Method section).
Line 221: highly specific: the colony formation assay is mainly showing proliferation and not specific to a certain cell type
→ We have revised the sentence according to your comment.(Lines 229-233)
The conclusion is not a conclusion but a methodological summary of the study. The last sentence is more a recommendation than a part of the conclusion.
→ We have revised the conclusion according to your comment (Lines 457-460).
Round 2
Reviewer 4 Report
The manuscript has significantly improved. It has been recognized that the cell population obtained by the selection method used, contains in majority fat precursors and a minority of muscle precursors. One has to read the entire manuscript thoroughly to pick that up though. A more pronounced section in the discussion and arguably the conclusion would be appropriate. The new text and data provided raise additional concerns and comments.
introduction: satellite cells between sarcolemma (not sarcomere) and BM
references 4-9 are missing from bibliography
line 247: it is appropriate to emphasize that only 11% of cells express CD56!
figure 2c,d: t-test not appropriate; use Oneway ANOVA instead. The qualification 'normal' is not clear, should be replaced by undifferentiated or control
figure 3C: PAX7 staining appears to be cytoplasmic instead of nuclear. Magnification still too low to really judge.
figure 5. It is very helpful to show the FTIR spectra. I am still sceptical about the specificity and therefore applicability to distinguish muscle from fat. It is immediately clear that it is incapable of ascertaining muscle differentiation. Whether it is capable of showing fat differentiation is not immediately clear either. In the ester graph differentiated muscle is lacking (so is differentiated fat in the amide graph, but here it is almost redundant as muscle diff is not clear anyway).
figure 6d. 10% apoptosis is appreciable! The level op apoptosis for fresh tissue is weird and probably artefactual (result of age of tissue).
Author Response
Dear Reviewer:
We thank you for providing valuable comment. We have revised the manuscript accordingly. Our responses to the comments are provided below.
Comments of Reviewer #4
The manuscript has significantly improved. It has been recognized that the cell population obtained by the selection method used, contains in majority fat precursors and a minority of muscle precursors. One has to read the entire manuscript thoroughly to pick that up though. A more pronounced section in the discussion and arguably the conclusion would be appropriate. The new text and data provided raise additional concerns and comments.
We thank you for your valuable comments. On behalf of all authors, I deeply respect your great input. We think these revisions have improved the manuscript substantially. 
introduction: satellite cells between sarcolemma (not sarcomere) and BM
→ We have revised the word in this sentence (Line 41).
references 4-9 are missing from bibliography
→ We have re-check the all references (Lines 512-526).
line 247: it is appropriate to emphasize that only 11% of cells express CD56!
→ We have added the sentence to explain accurately (Line 249).
figure 2c,d: t-test not appropriate; use Oneway ANOVA instead. The qualification 'normal' is not clear, should be replaced by undifferentiated or control
→ We have revised the statistical analysis section by providing more details (Lines 222-224, 295)
figure 3C: PAX7 staining appears to be cytoplasmic instead of nuclear. Magnification still too low to really judge.
→ We have added another high magnification images for Pax7 staining (Figure 3).  These data show that Pax7 (Green) clearly overlaps with nuclear staining (Blue).
figure 5. It is very helpful to show the FTIR spectra. I am still sceptical about the specificity and therefore applicability to distinguish muscle from fat. It is immediately clear that it is incapable of ascertaining muscle differentiation. Whether it is capable of showing fat differentiation is not immediately clear either. In the ester graph differentiated muscle is lacking (so is differentiated fat in the amide graph, but here it is almost redundant as muscle diff is not clear anyway).
→ Thank you for your comment. This time, we used this technology to analyze adipose differentiation and muscle differentiation in detail. However, quantifying the ability to differentiate in the spectrum is still in the developmental stage. Therefore, we moved these data to Supplement Figure 4. (Lines 327-334, 470-473)
figure 6d. 10% apoptosis is appreciable! The level op apoptosis for fresh tissue is weird and probably artefactual (result of age of tissue).
→ You're right. We think these cells include the influence of the process of experiment. We mentioned these data and move them for supplemental Figure 5. (Lines 350-351, 473-474)